# EFFICIENT SPARSE-WINOGRAD CONVOLUTIONAL NEURAL NETWORKS

**Xingyu Liu**[*], **Jeff Pool**[†], **Song Han**[‡§], **William J. Dally**[*†]
[*] Stanford University, [†] NVIDIA, [‡] Massachusetts Institute of Technology, [§] Google Brain
{xyl, dally}@stanford.edu

## ABSTRACT

Convolutional Neural Networks (CNNs) are computationally intensive, which limits their application on mobile devices. Their energy is dominated by the number of multiplies needed to perform the convolutions. Winograd's minimal filtering algorithm (Lavin, 2015) and network pruning (Han et al., 2015) can reduce the operation count, but these two methods cannot be directly combined – applying the Winograd transform fills in the sparsity in both the weights and the activations. We propose two modifications to Winograd-based CNNs to enable these methods to exploit sparsity. First, we move the ReLU operation into the Winograd domain to increase the sparsity of the transformed activations. Second, we prune the weights in the Winograd domain to exploit static weight sparsity. For models on CIFAR-10, CIFAR-100 and ImageNet datasets, our method reduces the number of multiplications by $10.4\times$, $6.8\times$ and $10.8\times$ respectively with loss of accuracy less than $0.1\%$, outperforming previous baselines by $2.0\times$-$3.0\times$. We also show that moving ReLU to the Winograd domain allows more aggressive pruning.

## 1 INTRODUCTION

Deep Convolutional Neural Networks (CNNs) have shown significant improvement in many machine learning applications. However, CNNs are compute-limited. Their performance is dominated by the number of multiplies needed to perform the convolutions. Moreover, the computational workload of CNNs continues to grow over time. LeCun et al. (1998) proposed a CNN model with less than $2.3 \times 10^7$ multiplies for handwritten digit classification. Later, Krizhevsky et al. (2012) developed AlexNet, an ImageNet-winning CNN with more than $1.1 \times 10^9$ multiplies. In 2014, ImageNet-winning and runner up CNNs increased the number of multiplies to $1.4 \times 10^9$ (Szegedy et al., 2015) and $1.6 \times 10^{10}$ (Simonyan & Zisserman, 2015) respectively. Despite the powerful representational ability of large scale CNNs, their computational workload prohibits deployment on mobile devices.

Two research directions have been explored to address the problem. Lavin (2015) proposed using Winograd's minimal filtering algorithm (Winograd, 1980) to reduce the number of multiplies needed to perform $3 \times 3$ kernel convolutions. On the other end, pruning the model (Han et al., 2015; 2016b) and exploiting the dynamic sparsity of activations due to ReLU also reduces the required multiplies.

Unfortunately, the above two directions are not compatible: the Winograd transformation fills in the zeros in both the weights and the activations (Figure 1(a)) – eliminating the gain from exploiting sparsity. Thus, for a pruned network, Winograd's algorithm actually *increases* the number of multiplies; the loss of sparsity more than offsets the reduced operation count.

In this paper, we introduce two modifications to the original Winograd-based convolution algorithm to eliminate this problem. First, we move the ReLU operation to be after the Winograd transform to also make the activations sparse at the point where the multiplies are performed. Second, we prune the weights after (rather than before) they are transformed. Thus, the weights are sparse when the element-wise multiply is performed — reducing the operation count. Together, these two modifications enable the gains of Winograd's algorithm and of exploiting sparsity to be combined. We open-source our code and models at `https://github.com/xingyul/Sparse-Winograd-CNN`.

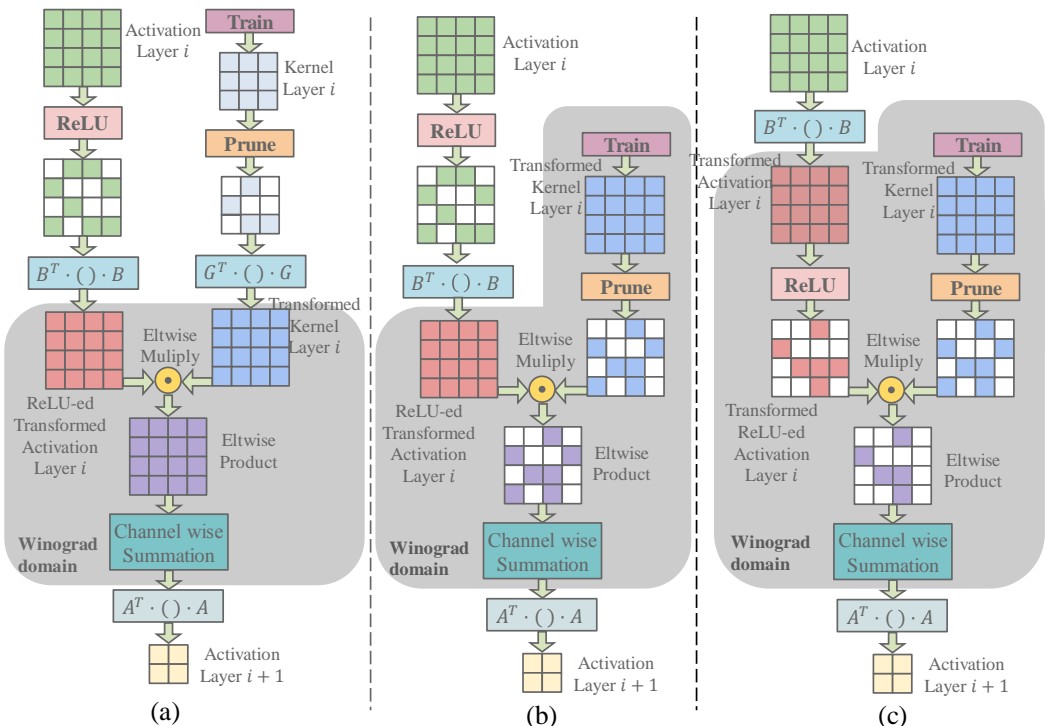

Figure 1: Combining Winograd convolution with sparse weights and activations. (a) Conventional Winograd-based convolution fills in the zeros in both the weights and activations. (b) Pruning the $4 \times 4$ transformed kernel restores sparsity to the weights. (c) Our proposed Winograd-ReLU CNN. Moving the ReLU layer after Winograd transformation also restores sparsity to the activations.

## 2  RELATED WORK

**Linear Algebra property in Convolution:** Previous research proposes using the linear algebra property of convolution to reduce the number of multiplies by trading additions for multiplies. Cong & Xiao (2014) convert convolution into matrix multiplies and utilize the linear algebra property at the sub-matrix block level. This approach achieves a 47% saving in multiplies. Lavin (2015) exploits the element-level linear algebra property of convolution, i.e. Winograd's minimal filtering algorithm (Winograd, 1980). This approach reduces the number of multiplies by $2.25\times$ to $4\times$, depending on the image patch size used in the algorithm. Winograd's algorithm is also used in a state-of-the-art deep learning library, cuDNN (Chetlur et al., 2014), to improve computation efficiency.

**Model Compression:** Model compression reduces the number of multiplies of CNNs by pruning network parameters (LeCun et al., 1990; Hassibi et al., 1993) and exploiting weight sparsity. Han et al. (2015; 2016b) proposed learning the sparsity pattern of network weights by eliminating weights whose absolute value is less than an empirical threshold. This approach can prune the convolutional layers of the model to only $30\% - 50\%$ of the original size and reduce the number of multiplies required. Liu et al. (2017) first proposed pruning and re-training the weights in Winograd domain for conventional Winograd convolution. Li et al. (2017) later showed promising results on large datasets and reported 90% sparsity in the Winograd parameters of AlexNet with less than $0.1\%$ accuracy loss.

**Dynamic Activation Sparsity:** The ReLU non-linearity sets activations whose values are negative to zero, causing dynamic sparsity in activations. Model compression can work in tandem with dynamic activation sparsity and reduce multiplication workload. Han et al. (2015) showed that exploiting sparsity of both weights and activations can reduce the number of multiplies by $4 - 11\times$. Huan et al. (2016) further proposed to manually set a small positive ReLU threshold at test time to exploit greater sparsity in activation without losing testing accuracy. Research in novel architectures also led to optimizations for deep learning accelerators to exploit the sparsity in activations. Han et al. (2016a) proposed using a Leading Non-zero Detection unit (LNZD) for their fully-connected layer accelerator to efficiently skip zeros in input activations. Albericio et al. (2016) proposed a similar mechanism for a convolution layer accelerator.

## 3 SPARSE WINOGRAD CONVOLUTION

We first introduce the conventional Winograd convolution and show how sparsity of weights or activations is lost during the dataflow of the algorithm. We then present the novel Winograd-ReLU CNN architecture. It preserves sparsity in both weights and activations before multiplies are performed and significantly reduces the computational workload.

### 3.1 SPARSITY IN CONVENTIONAL SPATIAL AND WINOGRAD CNN

The basic block of the conventional Winograd convolution algorithm works on an $p \times p$ patch (denoted by $d$) extracted with stride of $(p-2) \times (p-2)$ from an $H \times W$ input feature map. With "valid" padding, the $p \times p$ patch is convolved with a $3 \times 3$ kernel (denoted by $g$) to produce an $(p-2) \times (p-2)$ output patch (denoted by $S$). The output patches are assembled into an output feature map.

Input activation patch $d$ and kernel $g$ (spatial-domain activation and weights) are transformed using matrices $B$ and $G$ to be $B^T dB$ and $GgG^T$ (Winograd-domain activation and weights) respectively, both with shape $p \times p$. After element-wise product in Winograd-domain, the output activation $S$ is obtained using matrix $A$ (equation (1)). Matrices $B$, $G$ and $A$ are $p$-specific. When $p = 4$, $B$ and $A$ consists of 1, $-1$ and 0, so the multiplication with $B$ and $A$ only requires addition. It reduces the number of multiplies from $9(p-2)^2$ to $p^2$. Lavin (2015) gives details of the algorithm.

$$S = A^T[[GgG^T] \odot [B^T dB]]A \tag{1}$$

**Spatial Baseline Network:** When using a "vanilla" pruned network, as introduced by Han et al. (2015), a ReLU non-linear operation is performed by the previous layer on spatial-domain input $d$ and spatial-domain weight $g$ is pruned. The output activation patch $S$ is obtained from equation (2). This is illustrated in Figure 1(a) for $p = 4$. Though $g$ and $d$ may both be sparse due to pruning and ReLU respectively, the element-wise multiply is dense due to $G(\cdot)G^T$ and $B(\cdot)B^T$ transformations filling the spatial-domain zeros. Sparsity does not reduce the number of multiplies in Winograd's algorithm.

$$S = A^T[[G\text{Prune}(g)G^T] \odot [B^T\text{ReLU}(d)B]]A \tag{2}$$

**Winograd Native Pruned Network:** When using the Winograd-domain pruned network introduced by Liu et al. (2017) and Li et al. (2017), the spatial-domain input $d$ is ReLU-ed by the previous layer while the Winograd-domain weight $GgG^T$ is pruned. The output activation patch $S$ is obtained from equation (3). The algorithm when $p = 4$ is also illustrated in Figure 1(b). Though Winograd-domain weights are sparse due to pruning, Winograd-domain activations are still dense due to $B(\cdot)B^T$ transforms. The sparsity in spatial activations due to ReLU does not reduce the number of multiplies.

$$S = A^T[[\text{Prune}(GgG^T)] \odot [B^T\text{ReLU}(d)B]]A \tag{3}$$

### 3.2 WINOGRAD-RELU CNN

To address the above problems, we introduce the Winograd-ReLU Network. Instead of applying ReLU to the activations in the spatial domain, we apply ReLU to the activations in the Winograd domain, as in equation (4) and Figure 1(c). The ReLU operation zeros all negative *transformed* activations, reducing the number of multiplies in the Winograd domain.

$$S = A^T[[\text{Prune}(GgG^T)] \odot [\text{ReLU}(B^T dB)]]A \tag{4}$$

In the Winograd-ReLU CNN, we eliminate the spatial-domain kernel entirely. Because this ReLU is really associated with the previous layer, we perform this transformed ReLU starting with the second layer. We point out that the proposed new CNN architecture is *not* mathematically equivalent to the vanilla CNN nor the conventional Winograd CNN. Due to the change of network architecture, the training and pruning should also be changed. Our method operates in three phases: dense training, pruning, and retraining.

**Dense training:** we train a dense $p \times p$ kernel directly in the transform domain. The transformed kernel is initialized and trained directly by back-propagation through the inverse transform — eliminating the need to maintain a kernel in the spatial domain or to transform a spatial kernel.

**Pruning:** we prune the transformed kernel by computing the threshold $t$ required to achieve a desired pruning rate $r$ and setting all weights whose absolute value less than $t$ to zero. In our experiments, we used the same $r$ for all Winograd-ReLU layers. Because sensitivity varies from layer to layer, we expect that better performance could be achieved by varying the pruning rate $r_i$ for each layer $i$.

**Re-training:** we re-train the model using a "sparsity mask" to force the weights that were pruned to remain zero. The sparsity mask is computed during the pruning step and is kept constant during re-training. The gradient of the network's loss, $L$, with respect to the input activation and Winograd weights can be derived using the chain rule. Equation (5) shows the calculation of input activation gradient $\nabla_d L$ and Winograd weight gradient $\nabla_{GgG^T} L$ using the loss gradient passed from upstream layers $\nabla_S L$.

$$\nabla_{GgG^T} L = (A \nabla_S L A^T) \odot (B^T dB) \odot mask$$
$$\nabla_d L = B[(A \nabla_S L A^T) \odot (GgG^T) \odot mask]B^T \tag{5}$$

## 4 EXPERIMENTS

We applied the methodology described above to several different CNNs on different datasets. The original network models are chosen such that the majority of the convolution layers have $3 \times 3$ kernels. This ensures the largest portion of layers can be converted to Winograd convolution layers and ReLU be put in Winograd domain. We used image classification datasets of different scales: CIFAR-10, CIFAR-100 (Krizhevsky & Hinton, 2009) and ImageNet 2012 (Russakovsky et al., 2015). For network architectures, we chose VGG-nagadomi (Nagadomi, 2014), ConvPool-CNN-C model (Springenberg et al., 2015) and a variation of ResNet-18 (He et al., 2016a) respectively on three datasets. Using the Tensorflow (Abadi et al., 2016) framework, we trained the spatial baseline CNN, corresponding conventional Winograd CNN, and Winograd-ReLU CNN models from scratch. Then the three models are iteratively pruned and re-trained. For a specific dataset, we used the same data augmentation for the training of all models on the dataset.

### 4.1 CIFAR-10

We used VGG-nagadomi (Nagadomi, 2014) on the CIFAR-10 dataset. VGG-nagadomi is a lightweight version of VGGNet (Simonyan & Zisserman, 2015). It contains 8 convolution layers with $3 \times 3$ kernels. The best reported validation set accuracy it achieves on CIFAR-10 is $93.31\%$ (Nagadomi, 2014). We trained three models from scratch. The corresponding conventional Winograd CNN model and Winograd-ReLU CNN model can achieve validation set accuracy of $93.30\%$ and $93.43\%$ respectively. The first convolution layer is most sensitive to pruning and we set its density to a constant of $80\%$. We iteratively pruned and re-trained other convolution layers with density from $80\%$ down to $20\%$.

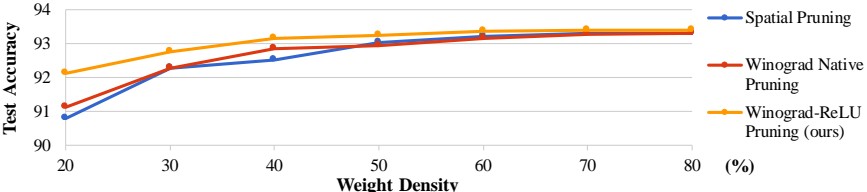

Figure 2: Test accuracy vs density for the three models in Figure 1 on VGG-nagadomi.

Figure 2 shows test accuracy as a function of weight density for the three models. The two baseline models can only be pruned to $60\%$ density before accuracy falls significantly ($> 0.1\%$). Our Winograd-ReLU CNN model can be pruned to $40\%$ density before falling to the same accuracy.

Table 1 shows the input activation density and compares the workloads for each pruned convolution layer in three models. Pruning two baseline models reduces the convolution layer workload by $5.1\times$ and $3.7\times$ [1] respectively. Pruning the Winograd-ReLU model reduces the convolution layer workload by $13.3\times$, a $2.6\times$ and $3.6\times$ improvement respectively over the two baselines. The improvement of overall network workload reduction is $2.2\times$ and $3.0\times$ respectively over two baselines.

---

[1] All Winograd CNN model workload reduction results include the intrinsic $2.25\times$ reduction.

Table 1: VGG-nagadomi weight and activation density on CIFAR-10.

| Layer | Spatial Baseline CNN Pruning (Han et al., 2015) | | | Winograd CNN Native Pruning (Li et al., 2017) | | | Winograd-ReLU CNN Pruning (**ours**) | | |
|---|---|---|---|---|---|---|---|---|---|
| | Density | | Workload | Density | | Workload | Density | | Workload |
| | Weight | Act | | Weight | Act | | Weight | Act | |
| conv0 | 80% | 100% | 80% | 80% | 100% | 80% | 80% | 100% | **80%** |
| conv1 | 60% | 50% | 30% | 60% | 100% | 27% | 40% | 46% | **8%** |
| conv2 | 60% | 19% | 12% | 60% | 100% | 27% | 40% | 39% | **7%** |
| conv3 | 60% | 37% | 22% | 60% | 100% | 27% | 40% | 40% | **7%** |
| conv4 | 60% | 18% | 11% | 60% | 100% | 27% | 40% | 40% | **7%** |
| conv5 | 60% | 26% | 15% | 60% | 100% | 27% | 40% | 38% | **7%** |
| conv6 | 60% | 24% | 14% | 60% | 100% | 27% | 40% | 35% | **6%** |
| conv7 | 60% | 35% | 21% | 60% | 100% | 27% | 40% | 36% | **6%** |
| conv total | - | - | 20%(5.1×) | - | - | 27%(3.7×) | - | - | **8%(13.3×)** |
| overall | - | - | 21%(4.7×) | - | - | 29%(3.5×) | - | - | **10%(10.4×)** |

## 4.2 CIFAR-100

We used the ConvPool-CNN-C (Springenberg et al., 2015) model on on the CIFAR-100 dataset. ConvPool-CNN-C contains 9 convolution layers, out of which 7 have $3 \times 3$ kernels. We trained three models from scratch. The spatial baseline CNN model and conventional Winograd CNN model can achieve single model validation accuracy of 69.34% and 69.32% respectively. The corresponding Winograd-ReLU network model can achieve validation set accuracy of 69.75%. We pruned the first convolution layer to a constant density of 80%. We iteratively pruned and re-trained the other layers to densities from 80% down to 20%.

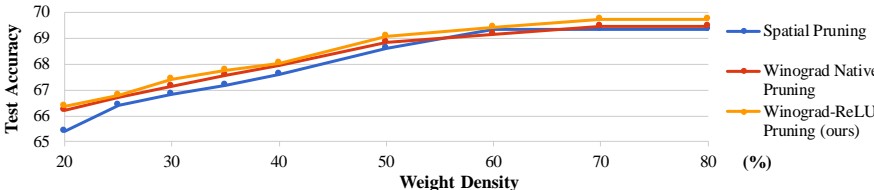

Figure 3: Test accuracy vs density for the three models in Figure 1 on ConvPool-CNN-C.

Figure 3 shows the accuracy as a function of density for spatial baseline and Winograd-ReLU models. The spatial-baseline and Winograd-ReLU models can be pruned to 60% density without significant ($> 0.1\%$) loss of accuracy. In contrast, the conventional Winograd CNN model can only be pruned to 70% density. At a given density, the Winograd-ReLU model has the highest accuracy.

Table 2: ConvPool-CNN-C weight and activation density on CIFAR-100.

| Layer | Spatial Baseline CNN Pruning (Han et al., 2015) | | | Winograd CNN Native Pruning (Li et al., 2017) | | | Winograd-ReLU CNN Pruning (**ours**) | | |
|---|---|---|---|---|---|---|---|---|---|
| | Density | | Workload | Density | | Workload | Density | | Workload |
| | Weight | Act | | Weight | Act | | Weight | Act | |
| conv0 | 80% | 100% | 80% | 80% | 100% | 80% | 80% | 100% | **80%** |
| conv1 | 60% | 53% | 33% | 70% | 100% | 31% | 60% | 54% | **14%** |
| conv2 | 60% | 52% | 32% | 70% | 100% | 31% | 60% | 53% | **14%** |
| conv3 | 60% | 77% | 46% | 70% | 100% | 31% | 60% | 54% | **14%** |
| conv4 | 60% | 35% | 21% | 70% | 100% | 31% | 60% | 54% | **14%** |
| conv5 | 60% | 32% | 19% | 70% | 100% | 31% | 60% | 42% | **11%** |
| conv6 | 60% | 56% | 33% | 70% | 100% | 31% | 60% | 43% | **11%** |
| conv total | - | - | 29%(3.5×) | - | - | 31%(3.2×) | - | - | **14%(7.1×)** |
| overall | - | - | 30%(3.4×) | - | - | 32%(3.1×) | - | - | **15%(6.8×)** |

Table 2 shows the input activation density and compares the workloads for each pruned convolution layer in three models. Pruning two baseline models reduces the convolution layer workload by $3.5\times$ and $3.2\times$ respectively. Pruning the Winograd-ReLU model reduces the workload by $7.1\times$, a $2.1\times$ and $2.2\times$ improvement respectively over the two baselines. The improvement of overall network workload reduction is $2.0\times$ and $2.2\times$ respectively over two baselines.

### 4.3 IMAGENET

We used a variation of the full pre-activation version (He et al., 2016b) of ResNet-18 (He et al., 2016a) on the ImageNet 2012 dataset. We used this version because it performs the best among various ResNet versions and its structure suits our Winograd-ReLU approach – its ReLU units are located before convolutions in the residual modules. The variation is different from original ResNet-18 by replacing all $2 \times 2$-stride $3 \times 3$ convolution layers with a $2 \times 2$ max-pooling layer followed by a $1 \times 1$-stride $3 \times 3$ convolution layer. Such difference ensure most of convolution layers can be converted to Winograd convolution layer. Another difference is that it doesn't have the last max pooling layer so the last group of residual modules has spatial size of $14 \times 14$, in order to keep the spatial size even instead of odd. This setting suits Winograd convolution with $p = 4$ best in that even spatial size is required for even $p$ values.

We trained three models from scratch. For single model and single central $224 \times 224$ cropping, the spatial baseline CNN model and conventional Winograd CNN model can achieve single model top-1/top-5 validation accuracy of $66.67\%/87.42\%$ and $66.84\%/87.47\%$. The corresponding Winograd-ReLU CNN model can achieve validation top-1/top-5 accuracy of $66.78\%/87.43\%$. We kept the first convolution layer intact. We iteratively pruned other convolution layers with density rate from $80\%$ down to $10\%$.

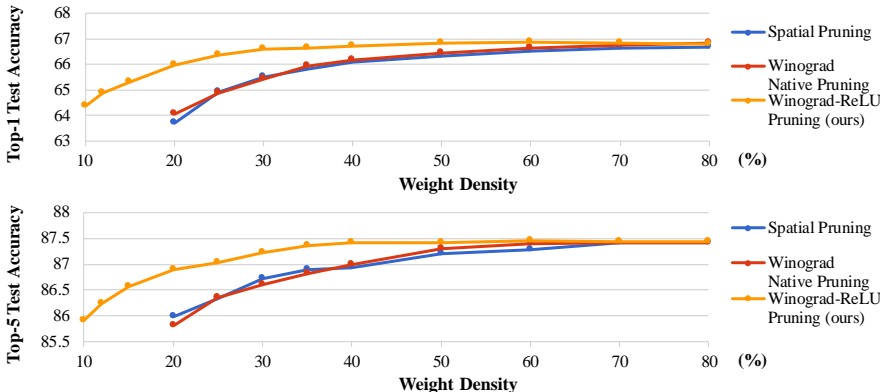

Figure 4: Top-1 and top-5 validation accuracy vs density for three models on a variation of ResNet-18.

Figure 4 shows the accuracy as a function of density for three models. The spatial baseline CNN model and conventional Winograd CNN model can be pruned to $60\%$ and $50\%$ respectively without significant ($> 0.1\%$) loss of top-1 or top-5 accuracy. The Winograd-ReLU model can be pruned much further, to $30\%/35\%$ density without significant ($> 0.1\%$) loss of top-1/top-5 accuracy. At these densities, top-1 accuracies are $66.53\%$, $66.45\%$ and $66.61\%$ for three models respectively, with a dense spatial baseline of $66.67\%$; top-5 accuracies are $87.29\%$, $87.30\%$ and $87.35\%$ for three models respectively, with a dense spatial baseline of $87.42\%$.

Table 3 shows the input activation density and compares the workloads for each pruned convolution layer in three models. Pruning the two baseline models reduces the convolution layer workload by $5.1\times$ and $4.5\times$ respectively. Pruning the Winograd-ReLU model reduces the workload by $13.2\times$, a $2.6\times$ and $2.9\times$ improvement respectively over the two baselines. The improvement of overall network workload reduction is $2.3\times$ and $2.6\times$ respectively over two baselines.

## 5 DISCUSSION

In this section, we summarize the experiment results and compare the three models in terms of a) weight and activation dimensions and b) the dynamic density of activations. We then visualize the kernels to illustrate the pattern of the proposed Winograd-ReLU model kernel.

Table 3: ResNet-18 variation weight and activation density on ImageNet.

| Layer | Spatial Baseline CNN Pruning (Han et al., 2015) | | | Winograd CNN Native Pruning (Li et al., 2017) | | | Winograd-ReLU CNN Pruning (**ours**) | | |
|---|---|---|---|---|---|---|---|---|---|
| | Density | | Workload | Density | | Workload | Density | | Workload |
| | Weight | Act | | Weight | Act | | Weight | Act | |
| res2a_2a | 60% | 90% | 54% | 50% | 100% | 22% | 35% | 48% | 8% |
| res2a_2b | 60% | 64% | 39% | 50% | 100% | 22% | 35% | 50% | 8% |
| res2b_2a | 60% | 71% | 43% | 50% | 100% | 22% | 35% | 50% | 8% |
| res2b_2b | 60% | 53% | 32% | 50% | 100% | 22% | 35% | 50% | 8% |
| res3a_2a | 60% | 94% | 56% | 50% | 100% | 22% | 35% | 49% | 8% |
| res3a_2b | 60% | 35% | 21% | 50% | 100% | 22% | 35% | 50% | 8% |
| res3b_2a | 60% | 47% | 28% | 50% | 100% | 22% | 35% | 49% | 8% |
| res3b_2b | 60% | 29% | 17% | 50% | 100% | 22% | 35% | 49% | 8% |
| res4a_2a | 60% | 88% | 53% | 50% | 100% | 22% | 35% | 49% | 8% |
| res4a_2b | 60% | 23% | 14% | 50% | 100% | 22% | 35% | 50% | 8% |
| res4b_2a | 60% | 36% | 22% | 50% | 100% | 22% | 35% | 50% | 8% |
| res4b_2b | 60% | 21% | 13% | 50% | 100% | 22% | 35% | 49% | 8% |
| res5a_2a | 60% | 45% | 27% | 50% | 100% | 22% | 35% | 50% | 8% |
| res5a_2b | 60% | 14% | 9% | 50% | 100% | 22% | 35% | 48% | 7% |
| res5b_2a | 60% | 16% | 10% | 50% | 100% | 22% | 35% | 48% | 7% |
| res5b_2b | 60% | 12% | 7% | 50% | 100% | 22% | 35% | 49% | 8% |
| conv total | - | - | 20%(5.1×) | - | - | 22%(4.5×) | - | - | 8%(**13.2×**) |
| overall | - | - | 21%(4.7×) | - | - | 24%(4.2×) | - | - | 9%(**10.8×**) |

## 5.1 Weight and Activation Dimension

In a convolutional neural network, a convolution-ReLU pair acts as a classifier on a spatial patch of an input feature. The dimension of the space being classified is the total number of elements passing through the ReLU layer. The decision boundaries of the classifier are determined by the weights. Insufficient non-zero weights or insufficient activations results in too simple a decision boundary and causes accuracy loss.

Experimental results have shown that Winograd-ReLU CNN can reach the same accuracy as both vanilla spatial baseline CNN and conventional Winograd CNN without pruning, and that Winograd-ReLU CNN is more robust to aggressive pruning. In this subsection we provide an explanation for the latter observation from the aspect of activation and weight dimensions. We provide a summary on dimensions in Table 4.

Table 4: Comparison of ReLU dimension and weight dimension in three types of networks. Assume the convolution-ReLU pair operates on input activation of spatial size of $H \times W$ and the number of input and output channels are $C$ and $K$ respectively.

| | Spatial Baseline CNN (Han et al., 2015) | Winograd native pruned CNN (Li et al., 2017) | Winograd-ReLU CNN (ours) |
|---|---|---|---|
| Weight dimension | $K \times C \times 3 \times 3$ | $K \times C \times p \times p$ | $K \times C \times p \times p$ |
| ReLU dimension | $H \times W \times K$ | $H \times W \times K$ | $\frac{p}{p-2}H \times \frac{p}{p-2}W \times K$ |

**Weight Dimension Increase:** Compared to a vanilla $3 \times 3$ CNN, a conventional Winograd CNN uses $(p \times p)$-dimension Winograd kernels. Training a Winograd CNN from scratch allows higher dimension $(p \times p)$ for Winograd kernels, and a Winograd-ReLU CNN shares these characteristics.

**ReLU Dimension Increase:** A major difference between our Winograd-ReLU CNN and conventional Winograd CNN is that the ReLU layers in Winograd-ReLU CNN have higher dimension. The dimension increase comes from the Winograd transformation extracting $p \times p$ feature patches with $(p - 2) \times (p - 2)$ strides from $H \times W$ activations. The total number of extracted Winograd-domain activations is $\frac{p}{p-2}H \times \frac{p}{p-2}W$, an increase from the spatial domain's $H \times W$.

We can see that our Winograd-ReLU architecture has an advantage on the dimensions of weights and activations over other two models. This means Winograd-ReLU CNNs classify on a higher dimension with more complex decision boundaries, which forms a stronger representational ability in high dimensional image feature space.

## 5.2 DYNAMIC ACTIVATION DENSITY

As is shown in the ImageNet results in the previous section, dynamic activation density of spatial baseline CNN model varies significantly among layers. Layers at earlier stages typically have higher density in activation than later stages. In Winograd-ReLU CNN model, the dynamic activation densities vary little among layers and are all close to $50\%$.

An explanation is that the nature of image convolution ensures activations $d$ to be *spatially* smooth. Thus, due to the structure of matrix $B$ (Lavin, 2015), 15 of 16 elements in the $4 \times 4$ matrix of Winograd-domain activation patch $B^T \cdot d \cdot B$ have a mean close to zero. This benefits classification within a patch since ReLU layer is most powerful when half of activations are positive.

## 5.3 KERNEL VISUALIZATION

We visualize the kernels of the proposed Winograd-ReLU model. We selected the first 6 input and output channels of layer res2a_2a of ResNet-18 at three different pruning densities. Unlike spatial domain kernels, Winograd-ReLU kernels do not show clear physical meanings such as edge or corner detectors. However, we observe that values of the $(2, 2)$ elements (from top-left, 1-based indices) in each kernel are typically distinct in a kernel and are most likely kept during aggressive pruning. A possible reason for this is that the $(2, 2)$ elements of Winograd-domain activation in a $4 \times 4$ patch are special: interested readers can calculate $B^T \cdot d \cdot B$ symbolically and will realize that $(2, 2)$ elements are the only elements that are transformed with a linear combination of only adding and no subtraction. In a spatially smooth activation patch, this means the $(2, 2)$ elements are the ones and the only ones with a non-zero mean.

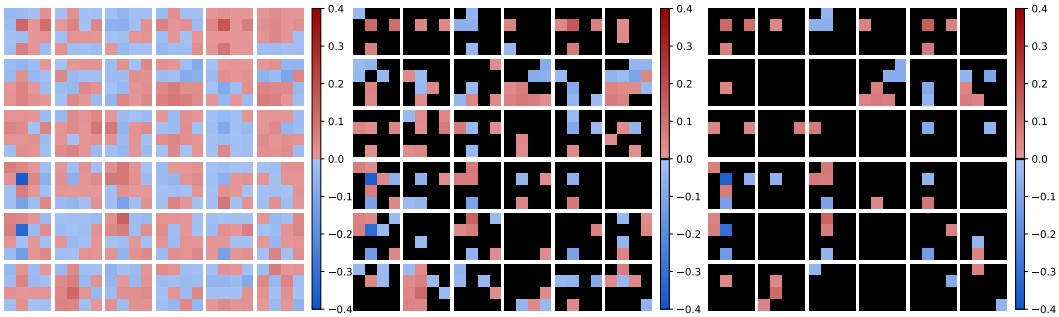

Figure 5: Kernels of ResNet-18 Winograd-ReLU model res2a_2a layer with density of $100\%$ (left, $87.43\%$ top-5 accuracy), $35\%$ (middle, $87.36\%$ top-5 accuracy) and $15\%$ (right, $86.57\%$ top-5 accuracy). Positive, negative and pruned weights are in red, blue and black respectively.

## 6 CONCLUSION AND FUTURE WORK

We have shown that we can combine the computational savings of sparse weights and activations with the savings of the Winograd transform by making two modifcations to conventional CNNs. To make the weights sparse at the point of multiplication, we train and prune the weights in the transform domain. This simple approach does not reduce the workload with respect to spatial pruning, though, so we move the ReLU non-linear operation after the Winograd transform to make the activations sparse at the point of multiplication. Moving ReLU to the Winograd domain also allows the weights to be more aggressively pruned without losing accuracy. With a $2 \times 2$ output patch ($p = 4$), the net result is a reduction of $10.4\times$, $6.8\times$ and $10.8\times$ in computation on three datasets: CIFAR-10, CIFAR-100 and ImageNet.

We plan to extend this work in the following directions. First, we expect that even greater savings on computation can be realized by using larger patch sizes (e.g., $p = 6$), and there may be benefit in exploring different Winograd transformation matrices ($B$, $G$ and $A$). Second, we expect that using different pruning rates $r_i$ for each network layer will help maintain accuracy and improve overall workload reduction. Finally, we expect that combining our Winograd-ReLU network with other network simplification techniques, e.g. quantization of weights and/or activations (Courbariaux et al., 2015; Lin et al., 2016; Rastegari et al., 2016), will reduce the energy of computation even further.

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
