# OpenReview forum: "Efficient Sparse-Winograd Convolutional Neural Networks"
_ICLR.cc/2018/Conference — Accept (Poster)_

### Official Review · AnonReviewer2 · 2017-11-25
**Good paper with thorough experiments**

**Rating:** 7
**Confidence:** 3

**Review:**

This paper proposes to combine Winograd transformation with sparsity to reduce the computation for deep convolutional neural network. Specifically, ReLU nonlinearity was moved after Winograd transformation to increase the dynamic sparsity in the Winograd domain, while an additional pruning on low magnitude weights and re-training procedure based on pruning is used to increase static sparsity of weights, which decreases computational demand. The resulting Winograd-ReLU
CNN shows strong performance in three scenarios (CIFAR10 with VGG, CIFAR100 with ConvPool-CNN-C, and ImageNEt with ResNet-18). The proposed method seems to improve over the two baseline approaches (Winograd and sparsity, respectively).

Overall, the paper is well-written and the experiments seems to be quite thorough and clear. Note that I am not an expert in this field and I might miss important references along this direction. I am leaving it to other reviewers to determine its novelty.

Putting ReLU in the Winograd domain (or any transformed domain, e.g., Fourier) seems to be an interesting idea, and deserves some further exploration. Also, I am curious about the performance after weight pruning but before retraining).

---

> ### Author Response · Authors · 2017-12-14
> **Response**
>
> Thanks for your comments.
> We agree that placing activation functions in other domains (e.g. Fourier) could hold more promise than we've uncovered so far.
> As far as accuracy before re-training, since we used iterative pruning and re-training, we provide top-5 accuracy drop for each pruning step of Winograd-ReLU CNN on ImageNet:
>
> original density | pruned density | original accuracy | pruned accuracy (without re-training)
> 100%                  |                           | 87.43%                 |
> 70%                    | 60%                   | 87.456%               | 87.338%
> 60%                    | 50%                   | 87.424%               | 87.202%
> 50%                    | 40%                   | 87.406%               | 86.672%
> 40%                    | 35%                   | 87.406%               | 86.784%
> 35%                    | 30%                   | 87.358%               | 86.286%
> 30%                    | 25%                   | 87.228%               | 85.692%
> 25%                    | 20%                   | 86.898%               | 84.466%
> 20%                    | 15%                   | 86.570%               | 80.430%
> 15%                    | 12%                   | 86.246%               | 79.246%
> 12%                    | 10%                   | 85.916%               | 77.128%

---

> > ### Comment · AnonReviewer2 · 2018-01-09
> > **Thanks for the response.**
> >
> > Thanks for the additional comments. I keep the rating.

---

### Official Review · AnonReviewer3 · 2017-11-28
**A promising Method, though with some limitations**

**Rating:** 7
**Confidence:** 4

**Review:**

This paper proposes a method to build a CNN in the Winograd domain, where weight pruning and ReLU can be applied in this domain to improve sparsity and reduce the number of multiplication. The resultant CNN can achieve ~10x theoretical speedup with little performance loss.

The paper is well-written. It provides a new way to combine the Winograd transformation and the threshold-based weight pruning strategy. Rather than strictly keeping the architecture of ordinary CNNs, the proposed method applied ReLU to the transform domain, which is interesting.

The results on Cifar-10 and ImageNet are promising. In particular, the pruned model in the Winograd domain performs comparably to the state-of-the-art dense neural networks and shows significant theoretical speedup.
The results on ImageNet using ResNet-18 architecture are also promising. However, no results are provided for deeper networks, so it is unclear how this method can benefit the computation of very deep neural networks

A general limitation of the proposed method is the network architecture inconsistency with the ordinary CNNs. Due to the location change of ReLUs, it is unclear how to transform a pretrained ordinary CNNs to the new architectures accurately. It seems training from scratch using the transformed architectures is the simplest solution.

The paper does not report the actual speedup in the wall clock time. The actual implementation is what matters in the end.

It will be more informative to present Figure 2,3,4 with respect to the workload in addition to the weight density.

---

> ### Author Response · Authors · 2017-12-14
> **Response**
>
> We appreciate your comments and questions; thank you. Let us address each in turn:
> 1) We agree, more work is warranted for deeper networks; we plan to explore this in the future.
> 2) It is true that the Winograd-ReLU CNN network architecture is not equivalent to an ordinary Winograd CNN. However, training a Winograd-ReLU network from scratch is a fairly simple solution. In fact there's no transformation from ordinary CNN weights to Winograd-ReLU CNN weights: the ReLU layer sizes are different. This cannot be compensated by any weight transformation.
> 3) While a reduction in wall clock time is the eventual goal, we focus here on a novel network type that reduces the theoretical number of operations needed, rather than the systems work needed to accelerate it.  This will need careful design with attention to architecture optimizations and tradeoffs, and we leave this as future work.
> 4) We'll try to find a clear way to present both density and workload in these figures, thanks for the suggestion.

---

> > ### Comment · AnonReviewer3 · 2017-12-22
> > **Thanks for the response**
> >
> > Thanks for the response. I hold a positive opinion on this paper.

---

### Official Review · AnonReviewer1 · 2017-11-30
**Well-written paper introducing a novel method of reducing multiplications in CNNs with very minor loss in accuracy**

**Rating:** 8
**Confidence:** 4

**Review:**

Summary:
The paper presents a modification of the Winograd convolution algorithm that enables a reduction of multiplications in a forward pass of 10.8x almost without loss of accuracy.
This modification combines the reduction of multiplications achieved by the Winograd convolution algorithm with weight pruning in the following way:
- weights are pruned after the Winograd transformation, to prevent the transformation from filling in zeros, thus preserving weight sparsity
- the ReLU activation function associated with the previous layer is applied to the Winograd transform of the input activations, not directly to the spatial-domain activations, also yielding sparse activations

This way sparse multiplication can be performed. Because this yields a network, which is not mathematically equivalent to a vanilla or Winograd CNN, the method goes through three stages: dense training, pruning and retraining. The authors highlight that a dimension increase in weights and ReLU activations provide a more powerful representation and that stable dynamic activation densities over layer depths benefit the representational power of ReLU layers.

Review:
The paper shows good results using the proposed method and the description is easy to follow. I particularly like Figure 1.
I only have a couple of questions/comments:
1) I’m not familiar with the term m-specific (“Matrices B, G and A are m-specific.”) and didn’t find anything that seemed related in a very quick google search. Maybe it would make sense to add at least an informal description.
2) Although small filters are the norm, you could add a note, describing up to what filter sizes this method is applicable. Or is it almost exactly the same as for general Winograd CNNs?
3) I think it would make sense to mention weight and activation quantization in the intro as well (even if you leave a combination with quantization for future work), e.g. Rastegari et al. (2016), Courbariaux et al. (2015) and Lin et al. (2015)
4) Figure 5 caption has a typo: “acrruacy”

References:
Courbariaux, Matthieu, Yoshua Bengio, and Jean-Pierre David. "Binaryconnect: Training deep neural networks with binary weights during propagations." In Advances in Neural Information Processing Systems, pp. 3123-3131. 2015.
Lin, Zhouhan, Matthieu Courbariaux, Roland Memisevic, and Yoshua Bengio. "Neural networks with few multiplications." arXiv preprint arXiv:1510.03009 (2015).
Rastegari, Mohammad, Vicente Ordonez, Joseph Redmon, and Ali Farhadi. "Xnor-net: Imagenet classification using binary convolutional neural networks." In European Conference on Computer Vision, pp. 525-542. Springer International Publishing, 2016.

---

> ### Author Response · Authors · 2017-12-14
> **Response**
>
> Thank you for your questions and comments; please allow us to address them here.
> 1) You are right to be confused - this should have been "p-specific," meaning the values of B, G, and A depend on p.  We'll correct this in a future version.
> 2) In general, our approach can be used wherever general Winograd convolutions can be used.  B, G, and A will be different for different patch sizes and filter sizes, and of course, we leave finding these and experimenting with larger sizes as future work.
> 3) Quantization approaches could fit well in the introduction; we'll try to find a way to make it clear that it may be orthogonal to pruning and Winograd convolutions.
> 4) Thanks for catching this typo.

---

> > ### Comment · AnonReviewer1 · 2018-01-12
> > **Good work**
> >
> > Thanks for the clarifications! I think the score is still appropriate.

---

### Author Response · Authors · 2018-01-29
**New Version**

We uploaded a new version of the paper. We also open-source our code at https://github.com/xingyul/Sparse-Winograd-CNN. The arXiv version is at https://arxiv.org/abs/1802.06367.

---

### Decision · Program_Chairs · 2018-01-29
**ICLR 2018 Conference Acceptance Decision**

**Decision:**

Accept (Poster)

**Comment:**

The paper presents a modification of the Winograd convolution algorithm that reduces the number of multiplications in a forward pass of a CNN with minimal loss of accuracy. The reviewers brought up the strong results, the readability of the paper, and the thoroughness of the experiments. One concern brought up was the applicability to deeper network structures. This was acknowledged by the authors to be a subject of future work. Another issue raised was the question of theoretical vs. actual speedup. Again, this was acknowledged by the authors to be an eventual goal but subject to further systems work and architecture optimizations. The reviewers were consistent in their support of the paper. I follow their recommendation: Accept.